# The Beneficial Interaction Between Human Well-Being and Natural Healthy Ecosystems: An Integrative Narrative Approach

**DOI:** 10.3390/ijerph22030427

**Published:** 2025-03-14

**Authors:** Natividad Buceta-Albillos, Esperanza Ayuga-Téllez

**Affiliations:** 1Organisation Engineering, Business Management and Statistics, Universidad Politécnica de Madrid, José Gutiérrez Abascal 2, 28006 Madrid, Spain; 2Buildings, Infrastructures and Projects for Rural and Environmental Engineering (BIPREE), Universidad Politécnica de Madrid, José Antonio Novais 10, 28040 Madrid, Spain; esperanza.ayuga@upm.es

**Keywords:** climate change, empathy, health, natural ecosystems, sustainability, well-being

## Abstract

This study highlights the lack of research on the relationship between ecosystem services, climate change, and human well-being. The experiences with the COVID-19 pandemic show the value of the natural environment for people’s well-being. We propose a framework that fosters an integrative approach to enhance our connection with nature, which is vital for tackling current environmental challenges. We reviewed over 70 articles and 160 references from databases such as Elsevier, ScienceDirect, Dialnet, MDPI, and Taylor & Francis, focusing on the correlation between pro-environmental behavior and emotional bonds with nature. Increasing our awareness of nature is crucial for fostering sustainable ecosystems. To deepen our understanding of how this connection influences human well-being and ecosystem health, we advocate for the application of specific neuroscience and artificial intelligence techniques. The study presents a compendium of prospective research topics for future investigation and analysis. In particular, it underscores the significance of this research for the development of effective policy and practical applications in the realm of conservation efforts.

## 1. Introduction

Forest ecosystems play a pivotal role in climate change adaptation and mitigation, absorbing approximately 50% of human carbon emissions and emissions reductions. In addition to providing us with various supporting, provisioning, regulatory, cultural, and aesthetic services that contribute to our well-being, they are essential for addressing social and ecological challenges, as they enable us to understand how environmental change and management decisions affect human well-being.

Adequate recognition and valuation of the benefits received from ecosystems is instrumental in fostering a comprehensive understanding of the significance of conserving and enhancing our natural environment. By prioritizing the well-being of ecosystems, we ensure our own and, by extension, that of future generations. The relationship we share with nature serves as the foundation for addressing our current environmental challenges.

A novel, diverse, and prolific field of study is currently developing that deals with the relationships between people and nature. This field includes what these relationships mean for the well-being of people and the care of ecosystems, seeking answers from the perspective of sustainability [1]. Prior research has demonstrated a positive correlation between pro-environmental conduct and the strength of the emotional bond with the natural environment [2].

Terrestrial and marine ecosystems are crucial for regulating the climate. They absorb approximately half the carbon emissions of human origin [3] and provide other support, provisioning, regulation, cultural, and aesthetic services that contribute to our well-being.

Adequately perceiving and valuing the benefits provided by ecosystems will help us understand the importance of conserving and improving our surroundings since by caring for our natural environment, we care for ourselves. Our own well-being and that of future generations depends on it. Previous studies have demonstrated that pro-environmental behavior is positively associated with the strength of the emotional bond with the natural world [4,5].

Conventional approaches fail to make social and ecological systems viable in the long term [6], to obtain the sustainability of ecosystems, and successfully address the problems that are being detected (lack of water, poverty, pandemics, climate change, loss of biodiversity, atmospheric pollution, etc.). Thus, it is necessary to address man’s relationships with nature from a more current perspective.

Some recent research after COVID-19 [1] revealed the importance that people assign to being in contact with nature and the benefits it represents for mental health, mindfulness, humility and empathy. Empathy in its broad sense refers to adopting the perspective of another and is a term that has evolved from different fields, with a current trend of empathy for the planet that highlights the interrelationships of collective “human and non-human” ecosystems [7]. Cultivating empathy and cohesion with humanity is positively related to pro-environmental commitment [8]. Connection with and empathy for nature is strongly associated with community cohesion, prosocial attitudes, and pro-environmental actions. Restoring human connection to nature can provide a pathway to promote the physical and spiritual well-being of individuals and communities as well as personal and social environmental responsibility [9].

Considering the increase in the urban population that has been occurring in recent decades, it is also necessary to prioritize nature in future urban design and development [10]. The design of homes based on the connection with nature is being widely studied [11]. Subjective well-being and the positive emotions that accompany it are the basis of a sustainable lifestyle [12,13].

Nature can have a positive influence on mental health, increase life satisfaction, and promote the development of positive personality traits, hence the need to be in contact with nature [8]. Natural environments are not only beneficial for human health but also for future sustainability and ecological diversity.

Our relationship with nature is the starting point from which to tackle our current environmental problems. Although we are starting to become aware of this and to recognize the inherent value of nature, there is still a long way to go before we adopt pro-environmental behaviors that can lead to a positive change that will benefit people, their communities, and the health of natural ecosystems. Following an ecosystemic approach, we conceive of people, their society, and their culture as an integrated part of ecosystems that link ecological and sociocultural systems together [14].

The aim of this study is to determine the current state of the art of the research on the relationship between the connection with nature and human well-being, pro-environmental behaviors, and health ecosystems; to build a constructive framework for looking at the positive relationship between ecosystem services and their impact on climate change and human well-being; and to identify the main gaps for future research.

The multidisciplinary nature of the subject matter renders an extensive state-of-the-art analysis essential. A range of philosophical, psychological, health, ecological, and sustainability aspects are employed to clarify the connections identified between well-being and nature. While some studies have demonstrated weak or absent correlations, a greater number of studies have shown a relationship to exist. The global confinement measures implemented in response to the pandemic provided a notable opportunity to examine people’s need for contact with nature, with research focusing on the changes it brought about.

## 2. Materials and Methods

The methodology of a narrative literature review requires selecting criteria to ensure the evaluation is comprehensive and impartial.

The following were necessary steps for carrying out the review: developing the selection criteria for a narrative literature review from the scope of this study, developing inclusion and exclusion criteria, employing a clear search strategy, and maintaining an iterative and flexible approach [15,16].

The scope for this review focused on a constructive framework for an integrative approach to the healthful relationship with nature as crucial starting point for addressing the current environmental challenges.

To develop search criteria for this critical topic, suitable keywords were identified to find the current literature in an iterative process.

For that, the search strategy for narrative review was designed following the process outlined. The Figure 1 shows the process followed in this study.

### 2.1. Criteria Review

The effective criteria selection involved identifying suitable keywords, conducting backward and forward searches to find the foundational and current literature, and minimizing manual screening [17]. The keywords associated with the research subject were identified, and the following keywords were chosen to use in the search for the information available on the topic study. They are listed below in alphabetical order: behavior, change, climate, cognitive, community, culture, development, diversity, ecosystems, education, emotions, empathy, environment, health, human, impact, landscape, nature, perception, positive, preoccupation, rural, setting, social, sustainability, systemic, territory, urban, and well-being.

The selection criteria considered studies allocated to interaction with nature, which found positive effects on health and well-being.

The selection criteria excluded the negative aspects and impact of the effect of contact with natural environments for the review topic (e.g., allergies, phobias, catastrophes, etc.) [18].

The search strategy was subsequently expanded to incorporate a string of terms including COVID-19, well-being, and contact with nature, taking into account the publications generated from 2020 onwards in which these terms appear simultaneously.

### 2.2. Open Search

In order to choose the appropriate databases and publication types to ensure the quality and reliability of the review, an Internet search in Google Scholar and indexing databases such as Scopus, Web of Science, and Dialnet was then performed with the selected keywords to locate the information sources and build the reference bibliography for this study.

The Internet search yielded the following results, with all the keywords selected from the last five decades shown in Figure 2.

Every decade, the number of published documents increased. The greatest increase occurred in the decade from 1990 to 1999 compared to the previous decade, with approximately six times more publications. In the rest of the decades, the increase doubled from one to the other.

In the last 4 years, the average number of annual publications increased. In the decade from 2010 to 2019, it reached a value of 85.5 annual publications, while in the 2020s, more than 110 annual publications were achieved; the average number of publications in the 4 years studied is 157.5.

### 2.3. Identification of Relevant Publications

The selection of articles comprised those published in peer-reviewed scientific journals, scientific books, and websites of official organizations, written in English and Spanish. The selection was made based on search words that appeared in the title, keywords, or abstract. In addition, reference lists were checked for other relevant articles. The most representative articles were selected according to the expertise in this field of the authors of the paper [19]. Of all these texts, those with the most relevance were compiled, which included those considered to be the most cited texts in recent bibliographies. Over 70 articles and 160 bibliographic references were selected as a result of this search.

The located articles were identified in the following information sources, ordered by number of articles found: Elsevier and ScienceDirect, Dialnet, MDPI, Taylor & Francis, and other sources. Elsevier, with its ScienceDirect search engine, stands out with 25% of the information collected for the study, followed by Dialnet, a multidisciplinary bibliographic search engine for Spanish-speaking scientific texts, with 19%. Figure 3 shows the main information sources identified in this study.

The collected articles correspond to the journals listed in Table 1, in order of highest to lowest number of articles found under study.

### 2.4. Cluster Classification

The cluster analysis as a segmentation aimed to identify homogeneous subgroups within the literature review. This process involved organizing information recollected into discrete classes to maximize within-group similarity and minimize among-group similarity, offering insights into information recovery to identify patterns of nature contact associated with well-being [20]. According to this exploratory analysis, the information was organized into five clusters:❖The benefits of nature;❖The attitude toward nature;❖The states of mind;❖The sociodemographic aspects;❖The methods used.

Every cluster is analyzed in the next section to obtain the main conclusions and to identify new areas of work and future practical applications.

As we advanced towards the study objective, we uncovered new information sources that broadened the scope and posed new challenges for the research area. This in turn requires the information to be expanded and reprocessed, which may ultimately lead to the reconfiguration and further redefinition of the research objective.

The final stage of the study involved the analysis of the five clusters that were identified for the purposes of the research. These results are presented in the following section.

## 3. Results and Discussion

The subsequent discourse herein delineates the analysis executed in each of the aforementioned clusters, along with the attendant outcomes and deliberations. In addition, an analysis of the research carried out on how COVID-19 has affected our relationships with the natural environment and a list of the gaps detected in the current state of knowledge are included.

### 3.1. From the Point of View of the Assessment of the Benefits of Nature

There is clear evidence in the earlier identified research of the value of natural spaces for people’s well-being. It is essential to be aware of nature in our lives to recover our essence and ensure the sustainability of the ecosystems on which we depend.

The concept of sustainable development was first described in the 1987 Brundtland Report as a development “that meets the needs of the present without compromising the ability of future generations to meet their own needs” [21]. The concept of sustainability—while retaining its initial significance—has gradually evolved since then to incorporate aspects such as poverty, equity, climate change, quality of life, and well-being. The challenges for a sustainable world are increasing every day and compel us to adopt healthy lifestyles.

The resources in the Healthy Parks, Healthy People program, which was launched in Australia in 1999 before spreading to several other countries with the support of the EUROPARC and IUCN workgroup (International Union for Conservation of Nature), highlight the benefits of nature for our health [22].

We present from the reviewed bibliography the main identified benefits of direct contact with nature.

Nature benefits physical health in the following ways:Favors recovery from illness [23];Decreases the heart rate [24,25];Reduces blood pressure [26];Produces vitamin D necessary for the organism [27];Favors the immune [28] endocrine, nervous, and respiratory systems [29];Reduces the risk of cardiovascular and other associated diseases [30].

Nature benefits psychological health in the following ways:Promotes self-esteem, autonomy, and a sense of responsibility [31];Favors reflection, concentration, and emotional intelligence [32];Is a source of stimulation for creativity [33];Influences positive emotions and emotional balance [34];Aids recovery from emotional fatigue, depression, states of anxiety, and stress [35];Restores attention [36].

Nature benefits social health in the following ways:Favors social relationships [37];;Generates a feeling of belonging to the community [38]Fosters and conserves local traditions and culture [39];Promotes curiosity about ecological knowledge, values, and sustainable attitudes [40].

Nevertheless, some research has shown uncertain or moderate effects on well-being [41,42,43,44,45,46], and certain studies have not obtained statistically significant results [47].

By integrating these benefits into the assessment of natural ecosystems and communicating them openly and transparently, we can raise society’s awareness of the vital importance of the natural environment. Figure 4 shows the link between healthy ecosystems and human well-being.

To build a society with increased human well-being, we need healthy natural ecosystems. We are both the solution and the problem in this society–nature relationship.

The physician and humanist Gregorio Marañón said in 1919 that “today’s hygiene requires an immediate derivation of citizens towards the countryside” [49], and more recently, Richard Louv suggested a natural healthcare system [50].

Today more than ever, we are becoming aware of the emergency of global warming and its repercussion on our lives. We now know how human activity impacts the climate and are searching for solutions so we can adapt our habits and mitigate the damage. There is no doubt that we have lived alienated from nature [51], disconnected from our own natural environment. The concept of alienation from nature refers to the growing disconnect between humans and the natural environment, a phenomenon that has significant implications for both individual well-being and environmental conservation. Addressing this disconnect requires a combination of promoting direct experiences with nature, rethinking educational practices, and adopting nuanced ethical and political frameworks. By fostering a deeper connection to nature, we can enhance individual well-being and support biodiversity conservation efforts [52,53].

We are an indivisible and essential part of nature. This approach recalls the concept of biophilia, which has its origins in the hypothesis of Kellert and Wilson, based on the natural relationship between human beings and nature as an innate biological need [54]. This perception of belonging to the natural world is increasingly being recognized in the current scientific literature [55,56,57].

The question is how to recover the connection with nature and reverse the phenomenon. We need to move from a polluting, greenhouse gas-emitting production system to a new and integrating model that regenerates the environment so that it absorbs the CO_2_ we emit, discarding the mechanistic vision and adopting a constructive, empathetic, and systemic approach. We are clearly seeing the start of a revolution in the concept of nature itself from something that is alien, egocentric, and to be exploited for our own ends to the idea that we are ourselves a part of nature. This shift can be seen in the books and reports published on this subject in recent years, the growing “re-naturalization” movement, and new nature-based solutions (NbS) [58].

Nature-based solutions (NbS) are defined as actions that are inspired from nature to address a range of societal challenges by leveraging natural processes, offering multifunctional benefits across environmental, social, and economic dimensions [59]. NbS offer a cost-effective approach to development, providing benefits such as flood mitigation, biodiversity conservation, and improved public health through exposure to natural environments. However, their success depends on careful planning, stakeholder involvement, and addressing existing challenges to fully realize their potential [60]. In Figure 5, we detail the conceptual framework of NbS.

NbS are strategies inspired by nature to address environmental challenges, such as climate change and biodiversity loss. They aim to create sustainable socio-ecological systems by integrating human well-being and environmental health [61]. This field explores how human interaction with nature can influence health, environmental behaviors, and the implementation of sustainable solutions to address ecological and societal challenges. NbS like the construction or protection of parks and green spaces can improve air quality, reduce urban heat, and promote physical activity, contributing to better health outcomes [62]. Holistic understandings of human–ecology–climate connections can enhance the effectiveness of NbS. There is a need for more robust, long-term studies to compare the effectiveness of NbS with non-NbS alternatives. The current literature shows biases towards urban environments and lacks comprehensive studies on the full range of well-being benefits from NbS [63].

This view contributes to an understanding of the systemic and dynamic changes through which everything is interconnected. In our Western cultural cosmovision [64], our belief in our own superiority over the rest of the natural world disconnects us from nature. This approach is losing ground, and in the new view of the ecosystem, we regenerate the natural environment; we “re-naturalize” ourselves. This idea of nature is also influenced by the current concept of sustainability, as seen in the Sustainable Development Goals (SDG) [65], which seek the best options for future generations based on the present moment, in a universal appeal to bring about a fairer, more inclusive, resilient, and sustainable world.

It is evident that after so many years of accelerated industrial development, new and invasive technologies, and urbanization as a lifestyle that alienates us from our natural environment, we have forgotten our essence and must re-establish our connection with nature. The negative consequences of being distant from the natural environment include attention deficit, hyperactivity, absence of creativity, sedentary habits, and obesity and isolation, known as nature deficit syndrome [28] or ecophobia [32]. Documented experiences with the COVID-19 pandemic have shown the role of green spaces in people’s well-being [66].

When people expand their identification to include their relationship with the natural environment—environmental identity [67]—they are more likely to act in an ecological way and transcend their unshakeable feeling of self. Some studies have identified the relationship between a connection with nature, concern for the environment, and sustainable attitudes [68]. This aspect is discussed within the deep ecology movement [69].

It is therefore important to highlight the value of the natural environment for people’s well-being and to adequately communicate this notion since unless something is adequately disseminated, it may as well not exist. Previous research based on the well-being sensitization model has demonstrated the positive relationship between a connection with nature and psychological well-being [70], although it is only significant for people who are emotionally in tune with the beauty of nature, regardless of age, sex, and personality. There is solid evidence to show that exposure to nature, such as walking in a natural setting, has positive psychological benefits [34]. If the concept of human well-being can be extended beyond the psychological dimension to include the physical, social, and others, this will be extremely useful in future research.

Using a systematic and transparent approach to measure ecosystem-based well-being can therefore reveal the state of human well-being within a framework of integrated ecosystem assessments [71], with the aim of achieving the mutual welfare of human beings and ecosystems [72].

A group of important international organizations has been working with the scientific community to develop an ecosystem accounting system as a tool for analyzing ecosystem regulation associated with the economy and other human activities [73] and incorporating key environmental information on natural consumable goods, residential flows, and environmental assets. Although several countries, such as Australia, the Netherlands, and the United Kingdom, have made some progress in accounting their ecosystems, there is still some way to go in measuring the state of ecosystem services and their costs and benefits.

### 3.2. From the Point of View of Man’s Attitude Toward Nature

More research on this subject is required to establish a link between connection and relationship with nature, the perception of natural beauty, and psychological well-being and a connection with nature and pro-environmental behaviors. The Natural Beauty Scale [74], which associates emotional and physiological awakenings in response to the perception of the beauty of the natural world, can be used to verify how people who perceive the beauty of nature benefit from its positive effects on their health.

The philosopher Wim Zweers proposed a series of possible models for the relationship with nature [75]:The despot: We manage nature for our own enjoyment;Enlightened ruler: We are guides and we collaborate with it;Nature’s steward: We care for nature, which belongs to its Creator;Partners with nature: We have equal conditions with it;Participants in nature: We are part of it;The mystic union between us and nature as one.

The deep ecology movement [76] promotes a paradigm shift in the relationship with nature, which involves a change in the value system. Subsequently, the value–belief–norm (VBN) theory [77] identifies three types of environmental concerns: egoistic, social–altruistic, and biospheric. Elsewhere, [78] developed a scale to measure environmental activities based on three defined profiles:Ecocentrism is the belief in the intrinsic importance of nature, which is positively correlated with nature conservation and environmental organizations and negatively related with environmental apathy; this represents the most advanced level of environmental concern and the highest level of moral reasoning. It is positively related with Kohlberg’s moral principle based on the concepts of justice, equity, rights, and obligations [79]. It is associated with the value–basis theory of environmental attitudes and the new ecological paradigm (NEP) [80] and negatively associated with the values of power and tradition [81];Anthropocentrism is the assignment of human qualities to nature [82] and the belief that nature is essential to human well-being [83]. This new era is identified as the Anthropocene due to global exploitation by humans [84];Apathy towards the environment describes people who are more likely to value the environment for its usefulness to humans [78], although this still remains to be demonstrated [85].

Notions of people–environment relationships such as the dominant social paradigm (DSP) [86] are in opposition to the new environmental paradigm (NEP) [80]. While the vision of the DSP focuses on individualism and is closely aligned with anthropocentrism, the NEP seeks harmony with nature to survive, a concept with links to ecocentrism. Both are frequently mentioned in the consulted literature.

The term dominant social paradigm (DSP) was first used by Pirages and Ehrlich, who described it as the “collection of norms, beliefs, values, habits, and so on that form the world view most commonly held within a culture” [87]. The DSP has evolved significantly over time, reflecting changes in societal values, technological advancements, and political ideologies [88]. The DSP continues to evolve, with a focus on sustainability, social justice, and technological innovation and the new challenges and opportunities of an ever-changing world. This paradigm is a framework for assessing quality of life and well-being as well as other domains, including values, the natural environment, and globalization [89]. Figure 6 shows a causal model of attitudes toward nature.

Our long-term survival requires us to adapt our individual and organizational behaviors to become environmentally sustainable [90]. Human beings’ relationship with nature is critical for tackling our modern environmental problems. The research supports the idea that as this relationship grows, empathy and willingness to collaborate also increase [91]. There is a need for research into how education programs that teach empathy, communication, and relational skills can improve sensitivity to the natural environment.

The relationship between connectivity with nature and ecological behavior is a significant area of research, focusing on how individuals’ connection to nature influences their environmental actions. Some research [92] found that the stronger the affective connection with the natural environment, the greater the intention to engage in environmental issues, but it is not so evident that environmental awareness leads to this type of behavior. One positive step forward in this area is the Nature Connectivity Scale (NCS) [93], as it shows a person’s connection with the environment and its relationship with ecological behavior, and this was also correlated with biospheric values [94]. The NCS promises to be a useful empirical tool for investigating people’s relationship with the natural world as a possible predictor of ecological behavior and subjective well-being; these aspects require an in-depth study and an analysis of the limiting conditions.

The specific mechanisms through which contact with nature and connectedness influence well-being and pro-environmental behaviors are not well understood. More research is needed to clarify these pathways and how they might be leveraged in interventions [95]. Continued research is necessary in this area, particularly with a focus on spatial dynamics and longitudinal studies to elucidate how different degrees of connectedness influence behavior. Understanding the spatial aspects of this connection can further inform policy and practice, highlighting the need for spatially explicit approaches in research and interventions [96].

One weakness of the current studies is that they cannot directly demonstrate the connection between objective self-awareness and pro-environmental attitudes. An objective for future research will therefore be to seek ways to measure the link between the NCS scale and the various personality characteristics to analyze the gap between awareness and environmental behavior. The studies designed to gain an understanding of this gap are based on sociodemographic surveys that produce somewhat unclear results and do not provide a definitive explanation, whereas very few research works have covered a beliefs and values model [77].

Thus, although numerous theoretical frameworks have been developed to explain the gap between the possession of environmental resources, knowledge, and environmental awareness and the display of pro-environmental behavior, there is no predictive model to explain this type of behavior. This suggests that the issue of pro-environmental behavior is so complex that it cannot be visualized through a single diagram. Following the pro-environmental behavioral model of Fliegenschnee and Schelakovsky [97], we list the main cognitive and emotional factors:Knowledge and environmental awareness;Social and personal norms;Incentives associated with environmental actions;Legislation and economic costs;Personal values.

### 3.3. From the Point of View of States of Mind

In psychology, a state of mind is seen as a dynamic and holistic construct that affects perception, attention, thought, affect, and behavior. It is determined by the balance between top-down and bottom-up processing, which provides a framework for understanding how these mental states operate and influence human cognition and behavior [98].

Stress is part of everyday life and can affect our attention and behavior. Stress has been shown to negatively affect real behavior by lowering the volume of donations to a foundation associated with climate change [99]. The research we consulted suggests that people are generally interested in climate change but less willing to pay attention to it when faced with other more immediate concerns.

Future research should analyze the influence of knowledge on cognitive perception and the effect of culture (values, norms, and beliefs) on the emotional perception of the natural environment [100,101,102]. According to Ulrich’s model [103], the first stage in the response is considered to be affective, requiring no cognitive processing and influencing cognitive perception and the final assessment of the environment. This analytical framework can be enriched with the decision architecture of Thaler [104], a Nobel Prize winner in economics, in what is known as behavioral economics. Aspects such as proximity to the natural world and convenience and access to the resources needed to facilitate sustainable behaviors may expand the pro-environmental behavioral model and act as a “nudge” to reduce the gap between awareness and behavior, although this still remains to be explored.

Several studies have addressed the association between mindfulness and the feeling of connection with nature and have verified how these associations vary significantly depending on the mindfulness measurement used. The Freiburg Mindfulness Inventory [105] captures aspects that are especially relevant to connectivity with nature, such as positive affect, life satisfaction, autonomy, and competence [106]. Other studies have demonstrated how connection with nature is related to positive emotions [34].

In the research on the link between greater mindfulness and connection to nature, [107] found that participants in a meditation session achieved a greater connection compared to participants in a control session.

Natural environments are particularly rich in the features required for restorative experiences. Attention restoration theory (ART) [108] analyzes the different types of experiences represented by recovery from directed attention fatigue, a key function in processing human information. Interaction with natural environments re-establishes the capacity to concentrate on a fixed task, improves effortless attention (bottom-up involuntary), and automatically favors positive feelings that can affect electrophysiological results [109]. Some neurophysiological laboratory experiments reported that natural environments stimulate the areas in the brain related to tranquility and relaxation [110,111].

It is therefore proposed that future studies should use an integrating framework to work on the long-term positive effects of the link between humans and the environment. The aim would be to study the physiological mechanisms underlying the restorative effects of the natural world and its characteristics as well as the aspects of mindfulness and connection with nature that are relevant to the interrelationship, in a cycle of reciprocal interactions between connection with nature and mindfulness.

The benefits of contact with nature are often mediated by psychological factors such as mood improvement and awe, which help shift attention away from negative self-focused thoughts. The perceived restorative quality of natural environments plays a crucial role in enhancing positive affect and reducing negative affect [18,112].

This review underscores the heterogeneity of the extant evidence pertaining to ART, with respect to the populations studied, the design of the studies, and the outcomes reported. A significant uncertainty persists with regard to the potential impact of exposure to natural environments on various aspects of attention [113].

Nature is often anthropomorphized in the environmental discourse and is attributed human qualities [114,115]. When people anthropomorphize nature, they tend to see nature itself and environmental problems as being closer and more understandable [82]. Future studies should explore whether anthropomorphizing nature impacts the way people relate to and behave with nature and whether it can lead to improvements in their well-being, by observing behavior in the laboratory and making real-life adjustments. Furthermore, research should seek to find the levers that activate an awareness of the natural environment and promote a feeling of connection to nature and the factors that influence an appreciation of the bond with nature and to identify the possible relationship between the bond with nature and pro-environmental behavior and—by extension—with life satisfaction. It should also analyze which cultures, attitudes, and personalities are pro-environmental and why.

This dimension of the analysis links with the role of empathy in nature conservation. Although empathy between people has been the object of numerous studies, this is not the case with empathy for nature. In recent years, some environmental researchers have studied empathy in the efforts to conserve the natural environment [116]. According to David Sobel, empathy should be taught from early childhood so that environmental education can serve as a basis in the future for the sustainable management of the environment. Improvements in efficacy can be achieved through experience-based education [117]. One transcendental experience in nature is worth more than thousands of facts [32].

Empathy for nature has been widely studied in a negative context in relation to how to mitigate environmental degradation [118]. In cases where the person feels themselves unable to change the situation, empathy may lead to empathetic anguish [119].

Empathy, defined in general terms as the ability to understand and share another person’s emotional experience [119,120], has often been considered by social scientists as the key to altruism and intergroup harmony [121,122]. Empathy in its two interrelated cognitive and affective components [120] drives helpful behaviors, which is why environmental researchers have come to believe that empathy for nature may be essential to conservation efforts. However, empathy is not the same as compassion. Emotional sharing is the key to defining the affective component of empathy. Various researchers have identified the role of emotions in the environment [92]. In fact, two of the six elements related to empathetic concern are described in terms of adjectives such as sympathetic, warm, compassionate, and tender. The anthropomorphizing of nature may therefore be an effective strategy for protecting the natural environment.

A comparison of empathy between humans and empathy for nature reveals that the psychological processes associated with empathy for people can also be applied to empathy for nature. The analyzed studies point out that it is difficult to have empathy for nature without developing empathy for other people. For example, one study [123] demonstrated how empathetic concern predicted concern for the biosphere. Some people consider themselves to be a part of nature [119,123], and for these individuals, the limit between humans and nature is blurred, so empathy for others implies empathy for nature and vice versa.

There is an urgent need to study how empathy for nature in a positive context is beneficial to us and can predict pro-environmental behavior. Individuals who feel connected to nature have more empathy for it and hence show greater pro-environmental behavior, and empathy for nature may improve the feeling of connection with nature. Future studies will need to demonstrate this bidirectionality.

The beneficial interaction between human well-being and healthy ecosystems underscores the need for integrated policy making and conservation efforts. By focusing on ecosystem-based development, sustainable land use, protection of critical natural assets, and the integration of natural capital into decision making, policies can effectively address the challenges of sustainability and climate change. These strategies are essential for maintaining ecosystem services and ensuring long-term human well-being [124].

### 3.4. From a Sociodemographic Point of View

Below, we present the results regarding the sociodemographic aspects of the consulted research consulted.

In terms of gender, women appear to have a more favorable relationship with the natural world [125]; they are more social and empathetic and also display more protective behavior toward nature, although it still remains to be demonstrated in future research whether there is really a clear difference from men. There is some inconsistency in the literature in this regard. Previous studies have found very few gender differences in the case of concern about pollution [126] and responsible environmental behaviors [127].

Grouped by age, older people appear to have a more developed environmental awareness than young people. Studies carried out with an older population have found a closer association with the natural world than in students, possibly because they have had more opportunities to relate to nature due to their age. More studies are required to confirm this in detail. Children perceive nature differently from adults, creating a living sense of continuity with the world around them and a basis for an empathetic relationship with the natural world, which is what makes us feel a responsibility to act and protect it [32].

Based on education level, the university population can be seen to be more pro-environmental than other young people and even more so in the case of biology students [85].

There is more environmental awareness in rural areas than in urban areas. Participants who have grown up in rural areas reported more positive inclinations to engage with the natural world than urban participants [128]. Local people also self-govern and have their own principles, standards, informal rules, and practices [129]. Improving country dwellers’ level of education and underlining the importance of ecosystem services could promote the protection of the natural world [130].

There have also been queries about the culture of the samples participating in the studies. It has been argued that countries whose citizens are pleasant and open tend to adopt more sustainable environmental measures or that empathy is more frequent in collectivist than in individualist cultures [131]. Nor is it clear whether Asian cultures are more pro-environmental than Western cultures, where consumption, industrialization, and urbanism have denatured the population. This points to the need for a more rigorous intercultural comparison.

In relation to personality type, an important objective for future research will be to identify the specific mechanisms through which the population’s personality characteristics can influence environmental sustainability at the local and national levels.

Other social and economic aspects, such as standard of living and income, were not identified in the consulted studies and could also be the subject of future research. Habits and lifestyles may be spontaneous activators of pro-environmental behavior [132].

It would be interesting in the future to analyze the perception of the environment, awareness, and environmental behavior of population samples based on lifestyle. Environmental perceptions are understood as the way each person appreciates and values their environment. This type of segmentation of the population may be useful for drawing practical conclusions and for gaining a better understanding of the pattern of environmental behavior and its relationship with the natural world in each lifestyle.

### 3.5. From the Point of View of the Assessment of the Methods Used

It is necessary to continue improving the use of quantitative and qualitative methodological instruments to analyze the link between people and the natural environment.

Overall, while there is substantial evidence supporting the positive effects of nature on health and well-being, inconsistencies in study designs and measures call for more rigorous research. Regular, intentional engagement with high-quality natural environments appears to be most beneficial [47,133].

The examination of the main methods used in the consulted bibliography, presented in Figure 7, clearly shows a significant number of research articles based on an analysis of the literature (52%), followed by 24% based on surveys. In most surveys, the environmental reference is addressed in a very general manner, and little attention is given to measuring pro-environmental behavior and the motives that explain it. We must emphasize the vital importance of posing clear and specific questions for the research objective.

Methods such as the application of neurosciences, eye tracking, focus groups, and scenario creation account for less than 1% of the consulted bibliography; these are including in the “Others” category, which comprises 11% of total methods.

It would be interesting to apply the research techniques enabled by telematic and scientific advances and the new ecophysiological, biotechnological, and statistical methods to leverage the advantages of artificial intelligence and social network analysis as well as the application of neurosciences in environmental perception, eye tracking, experiments in causal relationships, and the use of structural equations, to name a few.

In spite of these advances and the volume of data available, risks may result from the inconsistent use of databases, a lack of validation of the models built, and the absence of practical conclusions.

### 3.6. Results with COVID-19

The COVID-19 research emphasizes the how important it is for people to be in contact to nature.

The COVID-19 pandemic highlighted the essential role of urban and rural green spaces for societies facing a global public health crisis. There is a significant body of research devoted to analyzing the vital role of nature in human health and well-being [134,135].

Since the advent of the COVID-19 pandemic, there has been a substantial increase in the number of articles published annually, with a growth rate exceeding 84%. Specifically, the average number of articles published annually has increased from 85.5 in the 2010s (Figure 2) to 157.5 in the current decade.

Exposure to nature showed positive associations with physical activity level in the midst of COVID-19 confinement, and visiting green spaces and increasing garden use during confinement was associated with better self-rated physical health [135]. It was also observed that contact with nature and visits to green spaces can have a protective effect on mental health during public health crises. Access to gardens was shown to reduce the risk of depression and anxiety, and time spent in green spaces was demonstrated to reduce stress levels [136]. A study in Spain found that people turned to green spaces as a source of comfort, both directly and indirectly, to alleviate the negative effects of the pandemic [137]. A study in Bulgaria showed that visible access to vegetation around the house and neighborhood was associated with a decrease in symptoms of depression and anxiety during the pandemic [138].

Views of greenery from windows were found to provide micro-restorative episodes that aid in healing, psychological regeneration, and rehabilitation from traumatic events [139], including during confinement [140]. An investigation in Italy reported that greener views and access to private green spaces were associated with better mental health outcomes [138]. Other research in Egypt and Bangladesh showed that people in both countries increased time spent outdoors in green spaces after the period of confinement, and this increased time outdoors was associated with better mental health [141].

Physical activity offers important benefits for mental health and social well-being, but social distancing and quarantine requirements during the pandemic limited some people’s ability to engage in outdoor activities such as walking, running, or hiking. Walkable green spaces provide the opportunity for generating social connections. The use of accessible green space has the potential to enhance both physical activity as well as social cohesion, which can both positively affect mental health [142].

### 3.7. The Main Gaps

After a narrative review of the literature, the positive effects of nature on well-being were found to be well documented, but significant gaps remain in understanding the interplay of various nature-related factors, the role of non-material contributions, and the specific mechanisms at play. These gaps highlight areas where further investigation is needed to fully understand the complexities of how nature influences human behavior and well-being. Addressing these gaps could enhance the development of targeted interventions and policies to maximize the benefits of nature for diverse populations. These gaps identify a need to undertake new research to tackle the following areas by observing real behavior in the laboratory and/or making adjustments in real life and/or in virtual environments in different scenarios (urban, rural, protected spaces, stewardship of the territory, etc.).

The key gaps highlight areas where further investigation is needed to fully understand the complexities of how nature influences human behavior and well-being:There is a paucity of research examining the integration of individual nature-related factors such as connection to nature and time spent in nature. Furthermore, studies often fail to take into account the manner in which these elements interact to influence well-being [143];We need to better understand how human–nature relationships, such as cultural ecosystem services, contribute to well-being. Future research must focus on identifying the mechanisms that make people aware of the natural environment and connect them to it [144];New research is needed into how the restorative physiological mechanisms of natural environments are impacted by socio-economic factors. The effects of nature on mental health in diverse populations, including clinical groups, ethnically diverse communities, and low- and middle-income countries, have not been researched enough to date [133];Further research is required on the relationship between a connection with nature, well-being, pro-environmental behaviors, and life satisfaction. The mechanisms through which nature influences well-being and behaviors are not yet fully elucidated. Further research is necessary to clarify these pathways [96,145];There is a need for more robust, long-term studies to compare the effectiveness of NbS with non-NbS alternatives. The current literature shows biases towards urban environments and lacks comprehensive studies on the full range of well-being benefits from NbS [146];In order to establish a connection between environmental knowledge and awareness, it is crucial to consider proximity to the natural environment and availability of accessibly located resources for the adoption of pro-environmental behaviors. There is a need to integrate various theories to better understand the complex interplay factors [147];More research is required for the development of methodological tools, more sophisticated models, and frameworks that incorporate complex dynamics, social–cultural dimensions, and diverse environmental contexts. Causal inference methods and integrating quantitative and qualitative methods must be improved to enhance the effectiveness of strategies for managing human–environment interactions and promoting sustainability [148].

The complexity of natural ecosystems poses an important challenge for the development of empirical studies that help elucidate the person–ecosystem relationship.

## 4. Conclusions

This study reviewed a wide range of relevant research on the relationship with nature and the interactions between healthy natural ecosystems and human well-being, providing comprehensive insight and examining integrative trends, key findings, and gaps in the existing literature, particularly in the post-COVID-19 period. Several important implications for future research and practice in this topic are highlighted from the conducted analysis.

The bibliography we consulted clearly reveals a lack of studies that investigate natural environments from the positive standpoint of restoration. Most address the aggressive impact on the environment (fires, pollution, natural disasters, etc.), which generates environmental concern and leaves people feeling blocked due to their distance from the event, the lack of useful information to allow them to help, and the absence of training and the necessary resources. This situation gives rise to anxiety caused by an inability to act.

Nevertheless, most of the bibliography we analyzed highlights the importance of education and of raising awareness of the importance of caring for the environment in order to encourage the adoption of pro-environmental behaviors. In the COVID-19 situation, we are learning more about the great opportunity for recovering our relationship with nature for the well-being of people and the planet.

Several authors recognized that people who are connected with nature are more satisfied with their lives and report greater happiness and positive affect, social acceptance, mindfulness, and vitality. People who can perceive natural beauty tend to experience greater well-being. Tests carried out in previous studies designed to assess cognitive load and memory have found evidence of the capacity of natural environments to improve cognitive functions (e.g., cognitive control, restoration of attention, and concentration).

Human well-being and healthy ecosystems are crucial for sustainable development and climate change mitigation. Integrated policy making and conservation efforts are necessary. The integration of ecosystem-based development, sustainable land use, the protection of critical natural assets, and the incorporation of natural capital into decision-making processes is imperative for the effective management of sustainability and climate change challenges. These strategies are vital for the maintenance of ecosystem services and the assurance of long-term human well-being. The lack of quantitative relationship studies in this field is a key obstacle in understanding the complex processes and mechanisms between ecosystem services and human well-being.

Although numerous theoretical studies have sought to explain the gap between environmental resources, knowledge, and environmental awareness to demonstrate pro-environmental behavior, no definitive explanation has yet been found.

After a preliminary analysis of recent articles and the bibliography on the selected subject, we identified potential new research topics and recommend that future studies observe real behavior in the laboratory and/or make adjustments in real life and/or in virtual environments in different scenarios (urban, rural, protected spaces, stewardship of the territory, etc.).

In order to ensure the quality and validity of the results, it is essential that studies adopt a more rigorous approach to sample design. This will confirm that the results are meaningfully representative and provide a more robust basis for the conclusions drawn. Research in this field has many potential applications, including assessing the effectiveness of environmental education, sustainable forest management, community well-being, and climate change communication programs.

The complexity of natural ecosystems poses an important challenge for the development of empirical studies that help explain the person–ecosystem interaction.

## Figures and Tables

**Figure 1 ijerph-22-00427-f001:**
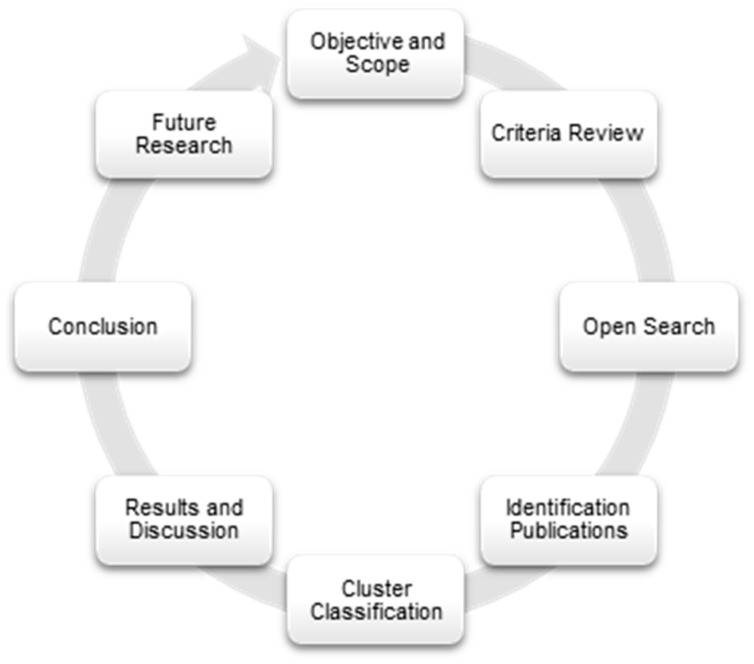
Process followed in this research. Prepared by the authors.

**Figure 2 ijerph-22-00427-f002:**
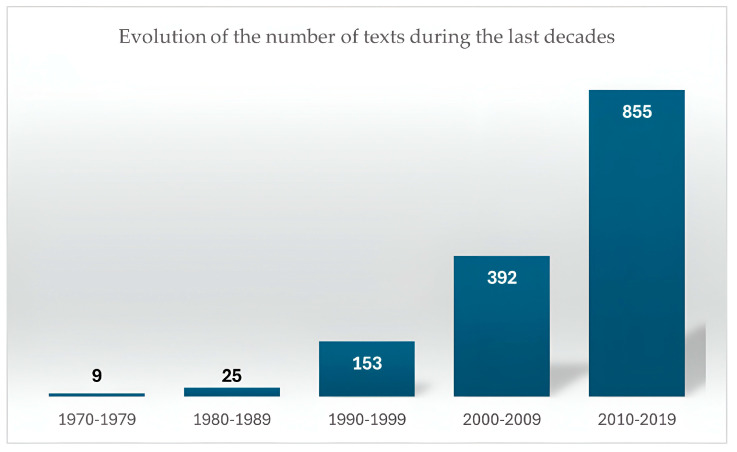
Number of texts (books and articles) obtained in the search in Google Scholar for the decades from 1970 to 2019. Prepared by the authors.

**Figure 3 ijerph-22-00427-f003:**
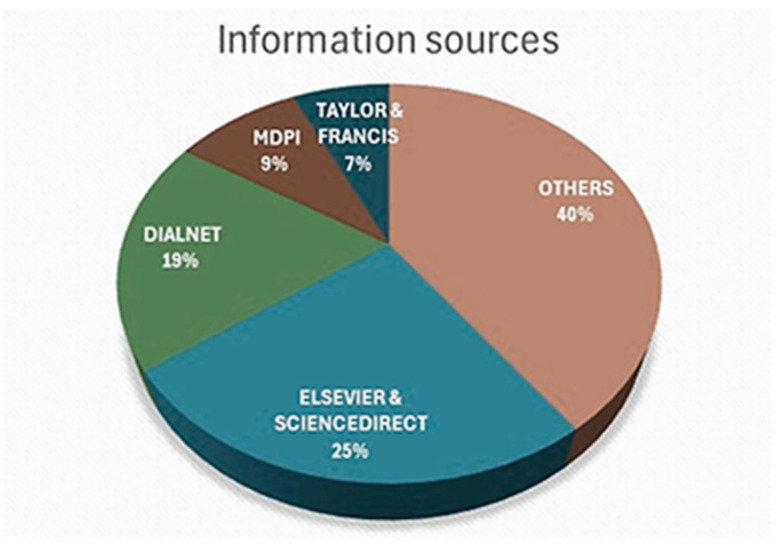
Sources of information identified in the search for open articles. Prepared by the authors.

**Figure 4 ijerph-22-00427-f004:**
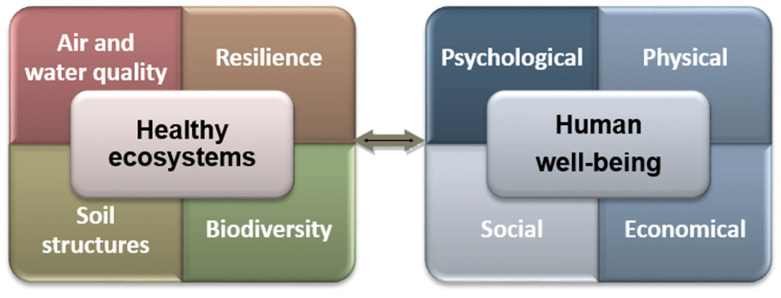
Relationship between healthy ecosystems and human well-being. Prepared by the authors based on the publication “Ecosystems and Human Well-being” (WHO, 2005) and other sources [48].

**Figure 5 ijerph-22-00427-f005:**
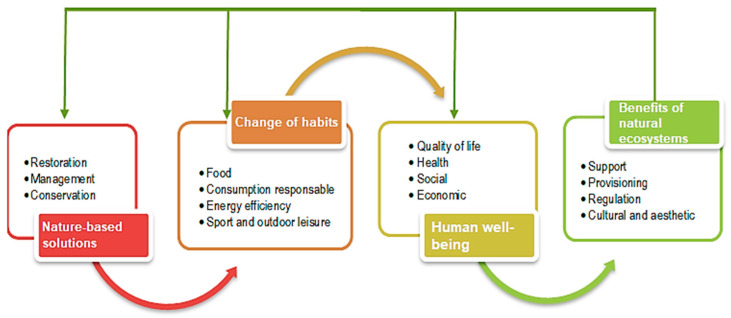
Prepared by the authors based on the IUCN’s conceptual framework of nature-based solutions (NbS) [58].

**Figure 6 ijerph-22-00427-f006:**
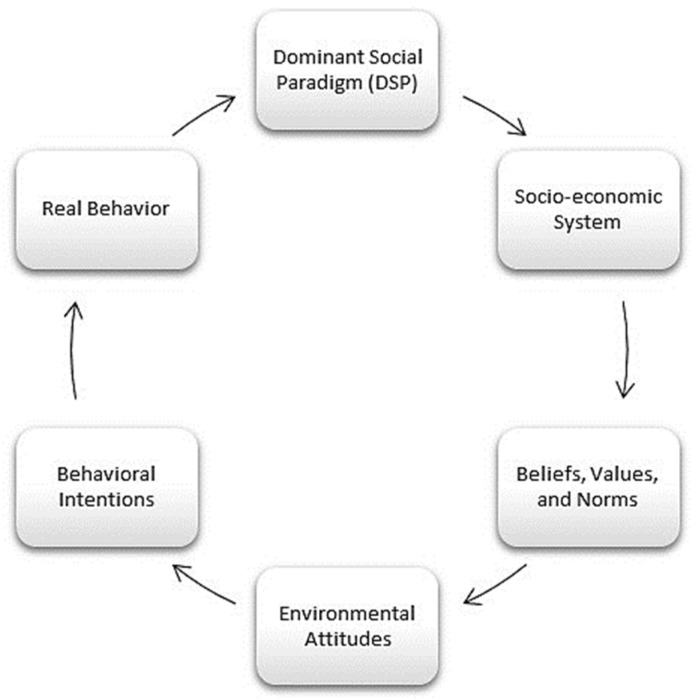
Causal model based on the DSP (Dominant Social Paradigm). Prepared by the authors based on [79].

**Figure 7 ijerph-22-00427-f007:**
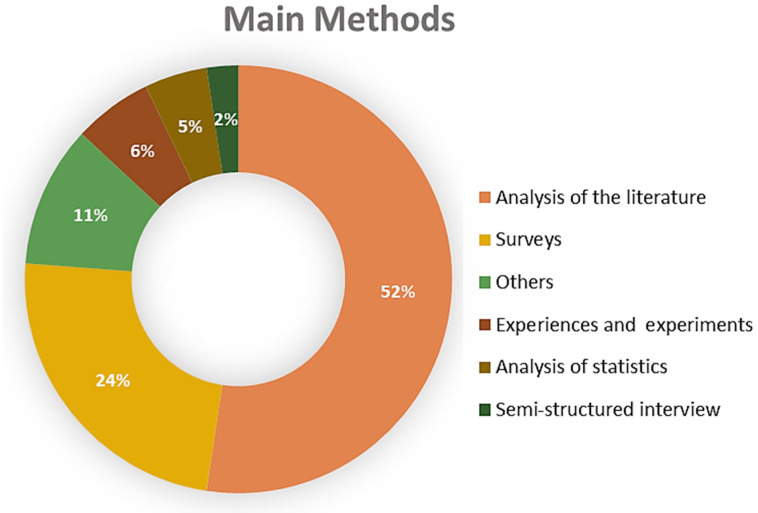
Methodology used in the bibliography consulted. Prepared by the authors.

**Table 1 ijerph-22-00427-t001:** Ranking of relevant consulted journals and their distribution. Own elaboration.

Ranking of Journals Consulted	Distribution (%)
Others	52%
Journal of Environmental Psychology	21%
Revista Investigaciones Geográficas	10%
Sustainability	8%
Ecosistemas. Revista Científica de Ecología y Medioambiente	5%
Journal of Ecosystem Health and Sustainability	4%

## Data Availability

No new data were created.

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
