# Peer review of "The Beneficial Interaction Between Human Well-Being and Natural Healthy Ecosystems: An Integrative Narrative Approach"

_ijerph, 2025, doi:10.3390/ijerph22030427_

Round 1
Reviewer 1 Report
Comments and Suggestions for Authors A beneficial interaction between human well-being and natural healthy ecosystems I have reviewed the above manuscript and in my opinion several points should be addressed before further consideration. These are as follows: 1. Abstract is generalized and should contain some data-oriented information. 2. Positive and negative influence of nature on mental health should be added in introduction citing latest research. 3. Objective and novelty of the study is not clearly presented in manuscript. Moreover, authors should discuss research gaps that are required to be addressed. 4. Methodology of the study is vague. Authors should elaborate the which established method has been used in synthesizing this manuscript. 5. There should be data on the number of articles considered or not considered for this review. 6. Authors should elaborate the selection criteria for considering literature for the review. 7. Authors have mentioned a list of health benefits of direct contact with nature. But, how this interaction can benefit health is not well discussed. 8. The role of green spaces in people’s wellbeing in context with COVID-19 pandemic should be discussed. 9. Authors should elaborate the Attention restoration theory in context with the human emotion and mental health. 10. There are some typos that should be corrected.Author Response
Thank you for your review and feedback. Your help is greatly appreciated and we hope it will improve our manuscript.
Following a thorough review of the abstract, objective and methodology, incorporating comments submitted, the following improvements have been made.
The work has been carried out from a positive, constructive, integrative framework of the relationship with nature.
The selection criteria for the literature review have been incorporated, and the main gaps found have been included.
The ART theory has been expanded, and a section dedicated to COVD-19 has been included.
Thirty-two additional references have been added, especially in the discussion.
The new texts incorporated are in blue in the paper.

Reviewer 2 Report
Comments and Suggestions for Authors Overall, this article focuses on the relationship between ecosystems and human well-being and is a meaningful review paper that I consider valuable. However, I still believe there are some details in this manuscript that need adjustment.- Lines 75-80 provide a detailed description of the keywords but do not specify the web databases or exact sources of information. The content later in the manuscript indicates that the primary search domain was Google Scholar. Does this mean that all articles and books indexed by Google Scholar were included in the research scope? Given the numerous keywords provided earlier, the number of potential sources could be much larger. It is recommended that the authors supplement the search strategy by specifying the search terms used.
- The results are mostly descriptive statistics. It is suggested that the authors add necessary data analyses (such as cluster analysis) to enhance the reliability of the paper’s results.
- Some figures and tables convey the same or similar information. It is recommended to merge them to make the manuscript more concise. For example, Table 1 and Figure 6 could be combined.
- The content in Section 3.4 on line 449 is too brief. It merely lists common research methods without detailed analysis of the necessity of selecting these methods and their contribution to the results. Additionally, this section is not reflected in the conclusion.
Author Response
Thank you for your review and feedback. Your help is greatly appreciated and we hope it will improve our manuscript.
Following a thorough review of the abstract, objective and methodology, incorporating comments submitted, the following improvements have been made.
The selection criteria for the literature review have been incorporated, and the cluster analyses done have been included.
Figures and tables have been reorganised and merged.
The results and discussion have been enhanced by the incorporation of new thirty-two additional references.
The new texts incorporated are in blue in the manuscript.

Reviewer 3 Report
Comments and Suggestions for Authors
In abstract:
- When mentioning "over 70 articles and 160 bibliographic references," it would enhance credibility to specify the criteria for selection. What specific themes or methodologies were prioritized?
- Instead of saying "other sources," consider naming a few additional databases or types of literature to provide a clearer scope of the research.
- The statement "Prior research has demonstrated a positive correlation between pro-environmental conduct and the strength of the emotional bond with the natural environment" would benefit from a citation or reference to specific studies to support this claim.
- The abstract ends somewhat abruptly. Consider adding a concluding statement that emphasizes the importance of this research for policy-making or practical applications in conservation efforts.
Revised Abstract Example
"This study highlights the lack of research on the relationship between ecosystem services, climate change, and human well-being. We propose a framework that fosters an integrative approach to enhance our connection with nature, which is vital for tackling current environmental challenges. We reviewed over 70 articles and 160 references from databases such as Elsevier, ScienceDirect, Dialnet, Mdpi, and Taylor & Francis, focusing on the correlation between pro-environmental behavior and emotional bonds with nature. Increasing our awareness of nature is crucial for fostering sustainable ecosystems. To deepen our understanding of how this connection influences human well-being and ecosystem health, we advocate for the application of specific neuroscience and artificial intelligence techniques."
Materials and Methods: The section could benefit from clearer headings and subheadings. For instance, consider using subheadings like "Keyword Identification" and "Information Search" to guide the reader through the methodology.
The phrase "following the process outlined here" could be improved by explicitly stating that it refers to Figure 1. For example: "following the process outlined in Figure 1."
Result:Consider breaking the text into clearer subsections with headings such as "Search Results," "Source Distribution," "Benefits of Nature," and "Assessment of Methods Used." This will help guide the reader through the various components of the methodology and findings.
-The transition between different topics could be smoother. For example, after discussing the search results, explicitly state that you will now discuss the benefits of nature before diving into that section.
-Ensure that all figures and tables are clearly referenced in the text.
- The statement "Over 70 articles and 160 bibliographic references were selected" could be clarified by specifying the criteria for selection. What made these articles relevant or significant?
-When discussing the increase in publications, it may be helpful to provide context for why this trend is important for the field.
- The mention of the Nature Connectivity Scale (NCS) and its potential should be expanded. How does it correlate with ecological behavior, and what implications does this have for future research?
- The conclusion should be revised by focusing on the important results of the reviewed studies.
Author Response
Thank you for your review and feedback. Your help is greatly appreciated and we hope it will improve our manuscript.
Following a thorough review of the abstract, objective and methodology, incorporating comments submitted, the following improvements have been made.
The methodology has been rearranged. The selection criteria for the literature review have been incorporated.
Figures and tables have been reorganised and merged.
The results and discussion have been enhanced by the incorporation of new thirty-two additional references and the main gaps found have been included.
The conclusions have been enriched with the main contributions of the study.
The new texts incorporated are in blue in the manuscript.

Reviewer 4 Report
Comments and Suggestions for Authors
(I also attach in a separate file.)
eview regarding the article about
„beneficial interaction between human well-behing and natural ecosystems“
First, I want to thank the authors for the interesting read which I did enjoy. Nevertheless I can not recommend the paper for publication, though I want to encourage a new submission after a thorough editing. While being an interesting read, the article seemed more like a political statement-paper, ignoring some scientific principles. The disregard of some major scientific principles is why I vote for „reject“ instead of „major revisions“. However, I do think it is possible to build something new from the work that you have already invested.
I will list details below, with the list not being complete but hopefully providing examples for each aspect.
-) First, the main goal of the paper is not completely clear. I do miss a specific research question and a specific research strategy. It is claimed the paper wants to „determine the current state of the art of the research on the relation between the connection with nature, ecosystem health and climate change“, but the paper is not marked as a „review“, be it systematic, scoping, narrative or else. (Given your focus, it probably fits best into the scheme of a narrative review. There are guidelines on how to conduct reviews that might be worth following, even if narrative reviews are less strict.)
-) Then, the method remains unclear. You claim you did an „open internet search“. But: Which search tools did you use? In figure 3 you mention google scholar in the figure notes, but was this the (only) basis for the literature search? The problem with search engines like google is that they can not distinguish between scientific peer-reviewed literature, websites, predatory (but seemingly good) journals and else. Thus, resulting literature can partly be problematic regarding its quality – an aspect I will also mention later.
Hence, please make traceable where you conducted your search.
You mention several (single) publisher websites as sources, aside from dialnet. However, when only looking at single publishers, you could get biased results, including only journals of the publishers you looked at anyway. (Plus I guess you miss Springer in your search, which though is a big player in the field as being the publisher of „nature“.)
What by the way is wrong with PubMed, Web of Science etc.? At least like this you make sure you only have listed and scientific journals.
-) Also, the list of keywords are somehow unclear. You list „urban“ as a keyword, but „urban“ alone might lead to you to completely wrong paths. Thus, I guess you mixed „urban“ with „urban greening“, „urban landscapes“ and so on. However, this is what you need to make traceable! Please provide an exact list of keywords so others – if they want – could redo exactly what you did and get the same results.
-) General traceability of methods: Aside from the fact that keywords and search strategies are not completely clear, there are several other aspects that need to be specified, e.g.: How did you identify the „most relevant“ articles? (Line 112). Most cited = according to whom? (Google scholar?) What is „recent“ bibliography? (Be specific – last 10 years? Last 5?).
What are „other sources“ of your literature? (Line 117).
How did you identify main ideas? (Line 130) – e.g. did you do a qualitative content analysis? Did you screen only titles for that, or abstracts, or conclusions, etc.? And if you did analyse that systematically, what was your classification criteria to identify the main ideas?
-) Mixture of results and discussion: This should clearly be avoided. It must be absolutely clear which statements are taken from literature and what the conclusions of the authors are.
If you want do it per sub-chapter instead of an overall discussion, that’s possible too, but there has to be a line between own interpretation and results from research used. Please pay attention that it is clear how you got to each statement.
Here are some examples:
Line 181f: „There is no doubt that we have lived alienated from nature“. (Says who? Also, this is not scientific language, see a little below for that too.)
Line 325f: „Future research should analyze the influence of knowledge on cognitive perception, and the effect of culture …“. How did you get to this conclusion? There is nothing it is based on before.
Or line 358f: „Nature is often anthropomorphized … and is attributed human qualities“. Who says that? Later you refer to one source (60), but it is not clear whether this statement is your conclusion or the one of source 60. (Plus, one source should not be „often“.)
-) The paper does not consider enough limits. This is a major issue! It might be due to a wrong search strategy (by also using a lot of grey literature), but the report of overall results is not critical enough. E.g., in terms of physical health (lines 149ff): a lot of benefits are reported, but recent reviews and meta-analyses also show that some benefits can not always be found - or are inconsistent, e.g. in respiratory health. Also, your article lacks a critical reflection of sometimes only very small effect sizes.
-) Be specific with terms! It is important that you make clear what you are talking about. As a POSITIVE example, you describe what you mean with empathy (line 46f resp. line 380ff – though in the latter it might be better to define it already before line 369).
There are many terms that you do not explain, but this would be of importance, so readers get a clear idea what you mean. E.g.: „state of mind“ (does that relate to well-being? Mental health?) Why not explain „objective self-awareness“? What do you mean by „industrial cosmovision“? (And, by the way, you use that term for a graph, but then never need it again.) etc.
-) Check for diligence: It is not called „Nature Connectivity Scale“ (NCS) but „Connectedness to Nature Scale“ (CNS), you even cite it yourself in the bibliography (source 68).
-) Mixture of topics: The mixture of subtopics is a little confusing to read. E.g.: How are lines 335-340 different from the chapter about attitudes? Why are beneficial effects on the „state of mind“ separated from „beneficial effects of nature“?
-) Extremely broad topic: Maybe due to the fact that you did not narrow down a specific research question, the topic you try to cover is extremely broad. It is hard to give an overview regarding so many things at once. E.g. it would be more than enough to „just“ focus on connections with phyiscal well being. Or only on demographic differences. Etc.
And then, even though the paper is already so broad anyway, ti still considers some things that are a little off topic, like in lines 55-59 (urban planning).
-) How much was found overall? I miss – especially as a result – an overview regarding how many articles actually dealt with which topic(s) or main idea(s). E.g.: 37% considered well-being (mostly – 80% - confirming positive relationships, while 20% found inconsistent results), 12% considered socio-demographic influence factors etc.
-) Scientific language and scientific language of interpretations: Please double-check for that.
E.g. regarding interpretations line 212: „The negative consequences of being distant from the natural environment include ….“ – These are not necessarily consequences. This is very likely based on correlational data, and causal pathways are likely, but not the only explanation! (E.g. it could also be another factor influencing both!)
Lines 202f: „our belief in our own superiority over the rest of the natural world disconnects us from nature“ – this is not scientific language.
Line 409ff: You state that there are gender differences, but then immediately add „although it still remains to be demonstrated whethere there is really a clear difference with men“ (which, I guess, also contains a grammar error). Things like that might be better be reformultated like „though not always reaching consistent results, majority of studies point towards the direction that women appear to show more pro-environmental behavior and report more connectedness with nature“ or similar. This also applies to line 274 (apathy) where you present a statement as given only to say it would still need to be demonstrated.
-) Literature sources: Some sources are excellent, others are not really scientific enough or not traceable. E.g. source 2 (citiation not complete): Why not use an IPCC source instead? Regarding this topic you might find one easily. Source 15: This too is incomplete citation I guess, it needs a website or else. But maybe you can find a more scientifc source too for that?
-) Minor remarks:
-) Why is the distribution to journals relevant? (E.g. figure 6) – please remove what’s irrelevant
-) Chapter 3.4 is numbered twice, once is in fact 3.5.
-) The wording above figure 10 is confusing, as you state there that the major method was a survey, when in the graph it was literature analyses. Maybe redefine as „aside from literature analyses, the major source was …“?.
-) Lines 33-36: I suggest to better delete this statement, as there are so many motivators to act pro-environmentally, and egoism is probably not the best example for it.
-) Line 44: Better rephrase sentences like this to „A recent study [1] highlights the importance that people give to the multiple values of nature…“ or such.
-) PS: Regarding the topic of your research, I guess you might also check the paper of Duong and Pensini, 2023 – maybe it provides something interesting for you.
OVERALL I think the most potential for working on your article is
a) to either use what you have and compare the „open internet search“ with a scientific search in order to see whether your main statements might be the same or whether you might be biased or
b) to split the paper you have in several papers (e.g. focusing only on socio-demographic factors etc.) and instead work on each part a little more critical.
But of course, there might also be other possibilities.
I do know that these were a lot of comments now. However, I do think the work that has already been invensted bears potential to overwork it and resubmit it, though I really want to highlight the need to follow principles of scientific writing (clear language etc.) and traceability.
Comments on the Quality of English Language
Please see in the file attached - most issues concern non-scientific English language, though I suggest to also check for minor grammar mistakes.
Author Response
Thank you for your review and feedback. Your help is greatly appreciated and we hope it will improve our manuscript.
Following a thorough review of the abstract, objective and methodology, incorporating comments submitted, the following corrections have been made.
The selection criteria for the literature review have been incorporated, and the main gaps found have been included.
Figures and tables have been reorganised and merged.
The results and discussion have been enhanced by the incorporation of new thirty-two additional references and the main gaps found have been included.
The NbS has been expanded, and a section dedicated to COVD-19 has been included.
Thirty-two additional references have been added, especially in the discussion.
The conclusions have been enriched with the main contributions of the study.
The new texts incorporated are in blue in the manuscript.

Reviewer 5 Report
Comments and Suggestions for Authors
1. Introduction is OK, but there is no additional background section or literature review. This would be good to add.
2. Line 44. "In the Study" is an odd way to reference an article. Maybe confuses with your own work too much.
3. Line 71. The aim is presented here, but this is all there there is that explains what you are doing in this paper. Please expand.
4. What do you mean an open internet search? Google? Why did you not use any of the common indexing databases, eg. scopus, web of science, etc.
5. Figure 1 and Figure 2 do not really add anything substantial to the paper.
6. Figure 3. It is a bit counter intuitive to have the most recent decades on the left and older on the right. Normally this would be the opposite, showing an increase, not a decrease.
7. Figure 5 and Table 1. Not sure how these are that relevant. Maybe if the aims of the paper are made more clear, these contributions make more sense, but as is, this entire first part of the results is not necessary since it doesn't discuss your research aim. Parts might be better suited in the methods.
8. The 'results' section itself once talking about the NBS is actually quite good. It is somewhat basic, without out much rigorous analysis and in many ways reads more like a textbook, so I question what the value of this really is, but it was interesting to read.
Author Response
Thank you for your review and feedback. Your help is greatly appreciated and we hope it will improve our manuscript.
Following a thorough review of the abstract, objective and methodology, incorporating comments submitted, the following corrections have been made.
The selection criteria for the literature review have been incorporated, and the main gaps found have been included.
Figures and tables have been reorganised and merged.
The results and discussion have been enhanced by the incorporation of new thirty-two additional references and the main gaps found have been included.
The NbS has been expanded, and a section dedicated to COVD-19 has been included.
Thirty-two additional references have been added, especially in the discussion.
The conclusions have been enriched with the main contributions of the study.
The new texts incorporated are in blue in the document.

Round 2
Reviewer 1 Report
Comments and Suggestions for Authors
Manuscript has been significantly improved.
Author Response
We would like to express our sincere gratitude for your review. Your comments and suggestions have been instrumental in enhancing the quality of the manuscript.
Reviewer 2 Report
Comments and Suggestions for Authors
The author did take care to revise and improve the quality of the manuscript. It would be better if the charts could be made more refined and scientific.
Author Response
We would like to express our sincere gratitude for your review. Your comments and suggestions have been instrumental in enhancing the quality of the manuscript.
We have undertaken a comprehensive revision of the paper, in addition to the figures and the table.
The new changes are in green text.
Reviewer 4 Report
Comments and Suggestions for Authors
Thank you for the additional work you put into your paper. As with the original article, it is obvious how much work and literature research you had to invest.
Nevertheless, I still can not recommend it for publishing without at least major changes, and I definitely cannot recommend publication under the “article”-category. However, a change into a narrative review or commentary seems possible, so given all the effort and the high importance of the topic I suggest a major revision (as review or commentary).
While some suggestions from the last round were accepted, some overall major problems still are not sorted out.
The most substantial one is probably that still limits are not considered enough. A big bonus goes to the subchapter of “main research gaps”, that was an excellent idea. However, what definitely also needs to be considered is that not all studies dealing with the welling-being and nature-interaction always find positive effects on health and well-being, or that effect sizes are sometimes really very small. E.g. in chapter 3.1 you list many studies that find positive effects, but you miss a list of studies that investigated the very same topics without being able to find (health-enhancing) connection. This, too, should be listed.
While the article selection is now explained, the search strategy still is not clear. It still is a random list of words in alphabetical order, and it is not clear how many of them had to be combined etc. – E.g., if I follow your description, I would get millions of results as EVERY article regarding “behaviour”, EVERY article regarding “culture” etc. would have to be screened or included, no matter how far from the topic or from which discipline. Thus, this still needs to be specified (e.g. was it a certain percentage of the words that had to apply? Or was it an absolute must that they ADDITIONALY include nature or green* or landscape or natur*? Etc.)
Please also check still for terms that might need explanation. I guess there was a misunderstanding in the first review round – it is not the usage of expressions like “state of mind” that is to be criticized but the absence of a definition as you are using it in your paper. Similar goes to the expression “aliented from nature”, though here I was not aiming for a lacking definition but the lacking traceability of the source (as it was not clear whether this is your conclusion or whether you cite other sources – it was the latter, so I suggest adding the references).
What is NbS? You never explain the abbreviation (Lines 256ff.), though you use it a lot. (I know, it’s nature-based solutions, but you never name it.)
Line 261: “NbS like parks …” rather: “NbS like the construction or protection of parks and green spaces…”.
The now-added chapter about COVID needs to be explained too, as it is not covered with the words of the search strategies, especially if you systematically looked for new literature. (E.g. mention that it was added “ex post” or similar, or that the search strategy later was additionally expanded for COVID or such.)
Overall, the article still seems too broad, with then again a lot of repetitions, as several chapters of course overlap. With an additional chapter now (regarding COVID, maybe a reviewer suggestion) it is even broader. – You could actually even consider breaking your paper into several parts and publish them separately. (Just an idea, not mandatory.)
Additionally:
Please also check for my extensive comments in the first round. E.g., still the term “industrial cosmovision” in figure 6 is never explained etc. It is still not explained why the publishing house “Springer” was not included etc.
Also, I would still recommend to clearly distinguish between results and discussion and not mix it together.
Line 589: Fist you talk about physical activity, then about social cohesion without a linkage between them. Do you mean that green space enhances both physical activity as well as social cohesion, which can both positively affect mental health? In this case please (re-)formulate explicitly, so the linkage becomes clear.
Minor notes:
There are still some diligence errors. E.g. figure 1 contains multiple typos.
Line 90, last word: You probably mean “research”, not “researches”?
Lines 101: “It is essential to decide on which types of study designs to searches”. – There is something wrong with this sentence…
Table 1: There is a line missing on the bottom of the table.
Line 156: “by number of articles found under study., listed in Table 1” – there is something wrong.
Line 342: “People who value the environment for its usefulness for humans are more likely to be apathetic, although this remains to be demonstrated”. – If it is not demonstrated yet, please change “are” to “could be”. If it was demonstrated in one study, but not found in another, rephrase the second part (“although contradictory results were found too” or such).
Line 565: There is a full stop missing at the end of the sentence.
You have two chapters named “conclusions”.
Author Response
We would like to express our sincere gratitude for your review. Your comments and suggestions have been instrumental in enhancing the quality of the manuscript.
We have undertaken a comprehensive revision of the paper, in addition to the figures and the table.
The new changes are in green text.
Thank you for the additional work you put into your paper. As with the original article, it is obvious how much work and literature research you had to invest.
Nevertheless, I still can not recommend it for publishing without at least major changes, and I definitely cannot recommend publication under the “article”-category. However, a change into a narrative review or commentary seems possible, so given all the effort and the high importance of the topic I suggest a major revision (as review or commentary).
Reply: Thank you for your review and important input. Indeed, changes have been made that allow the article to be seen as an analysis on the current state of research on the subject. We have completed the title of the article accordingly.
While some suggestions from the last round were accepted, some overall major problems still are not sorted out.
The most substantial one is probably that still limits are not considered enough. A big bonus goes to the subchapter of “main research gaps”, that was an excellent idea. However, what definitely also needs to be considered is that not all studies dealing with the welling-being and nature-interaction always find positive effects on health and well-being, or that effect sizes are sometimes really very small. E.g. in chapter 3.1 you list many studies that find positive effects, but you miss a list of studies that investigated the very same topics without being able to find (health-enhancing) connection. This, too, should be listed.
Reply: This has been included at the end of the section 3.1.
While the article selection is now explained, the search strategy still is not clear. It still is a random list of words in alphabetical order, and it is not clear how many of them had to be combined etc. – E.g., if I follow your description, I would get millions of results as EVERY article regarding “behaviour”, EVERY article regarding “culture” etc. would have to be screened or included, no matter how far from the topic or from which discipline. Thus, this still needs to be specified (e.g. was it a certain percentage of the words that had to apply? Or was it an absolute must that they ADDITIONALY include nature or green* or landscape or natur*? Etc.)
Reply: The methodology has been extended in this regard.
Please also check still for terms that might need explanation. I guess there was a misunderstanding in the first review round – it is not the usage of expressions like “state of mind” that is to be criticized but the absence of a definition as you are using it in your paper. Similar goes to the expression “aliented from nature”, though here I was not aiming for a lacking definition but the lacking traceability of the source (as it was not clear whether this is your conclusion or whether you cite other sources – it was the latter, so I suggest adding the references).
Reply: We appreciate your indication and the necessary references have been included in the revised text.
What is NbS? You never explain the abbreviation (Lines 256ff.), though you use it a lot. (I know, it’s nature-based solutions, but you never name it.)
Line 261: “NbS like parks …” rather: “NbS like the construction or protection of parks and green spaces…”.
Reply: The requested explanations have been included and the change in the line 261 has been made.
The now-added chapter about COVID needs to be explained too, as it is not covered with the words of the search strategies, especially if you systematically looked for new literature. (E.g. mention that it was added “ex post” or similar, or that the search strategy later was additionally expanded for COVID or such.)
Overall, the article still seems too broad, with then again a lot of repetitions, as several chapters of course overlap. With an additional chapter now (regarding COVID, maybe a reviewer suggestion) it is even broader. – You could actually even consider breaking your paper into several parts and publish them separately. (Just an idea, not mandatory.)
Reply: A justification of the importance of COVID 19 in this research has been included both at the end of the introduction and in section 3.6. In the methodology, a paragraph has been added at the end of 2.1 containing how the results of section 3.6 have been obtained.
In the introduction, a final paragraph has been added to justify the scope of this research.
Additionally:
Please also check for my extensive comments in the first round. E.g., still the term “industrial cosmovision” in figure 6 is never explained etc. It is still not explained why the publishing house “Springer” was not included etc.
Reply: Of course in the repositories of Scopus, Schoolar Google, etc. the journals of this publisher are included. The publisher “Springer” was included in the section on “Others” as we have not found many references to works by this publisher.
Also, I would still recommend to clearly distinguish between results and discussion and not mix it together.
Reply: Thank you for your recommendation. However, in consideration of the nature of the research as a state-of-the-art analysis in an integrative narrative approach, it appears more appropriate to introduce the commentary discussion concurrently with the presentation of the results.
Line 589: Fist you talk about physical activity, then about social cohesion without a linkage between them. Do you mean that green space enhances both physical activity as well as social cohesion, which can both positively affect mental health? In this case please (re-)formulate explicitly, so the linkage becomes clear.
Minor notes:
There are still some diligence errors. E.g. figure 1 contains multiple typos.
Line 90, last word: You probably mean “research”, not “researches”?
Lines 101: “It is essential to decide on which types of study designs to searches”. – There is something wrong with this sentence…
Table 1: There is a line missing on the bottom of the table.
Line 156: “by number of articles found under study., listed in Table 1” – there is something wrong.
Line 342: “People who value the environment for its usefulness for humans are more likely to be apathetic, although this remains to be demonstrated”. – If it is not demonstrated yet, please change “are” to “could be”. If it was demonstrated in one study, but not found in another, rephrase the second part (“although contradictory results were found too” or such).
Line 565: There is a full stop missing at the end of the sentence.
You have two chapters named “conclusions”.
Reply: All these corrections have been made.